# Anomalous formation of trihydrogen cations from water on nanoparticles

M. Said Alghabra[1], Rami Ali[1,2], Vyacheslav Kim[1], Mazhar Iqbal[1], Philipp Rosenberger [3,4], Sambit Mitra[3,4], Ritika Dagar[3,4], Philipp Rupp[3], Boris Bergues[3,4], Deepak Mathur[5], Matthias F. Kling [3,4] & Ali S. Alnaser [1✉]

Regarded as the most important ion in interstellar chemistry, the trihydrogen cation, $H_3^+$, plays a vital role in the formation of water and many complex organic molecules believed to be responsible for life in our universe. Apart from traditional plasma discharges, recent laboratory studies have focused on forming the trihydrogen cation from large organic molecules during their interactions with intense radiation and charged particles. In contrast, we present results on forming $H_3^+$ from bimolecular reactions that involve only an inorganic molecule, namely water, without the presence of any organic molecules to facilitate its formation. This generation of $H_3^+$ is enabled by "engineering" a suitable reaction environment comprising water-covered silica nanoparticles exposed to intense, femtosecond laser pulses. Similar, naturally-occurring, environments might exist in astrophysical settings where hydrated nanometer-sized dust particles are impacted by cosmic rays of charged particles or solar wind ions. Our results are a clear manifestation of how aerosolized nanoparticles in intense femtosecond laser fields can serve as a catalysts that enable exotic molecular entities to be produced via non-traditional routes.

[1] Department of Physics, American University of Sharjah, Sharjah, UAE. [2] Department of Physics, The University of Jordan, Amman, Jordan. [3] Department of Physics, Ludwig-Maximilians-Universität Munich, Garching, Germany. [4] Max Planck Institute of Quantum Optics, Garching, Germany. [5] Department of Atomic and Molecular Physics, Manipal Academy of Higher Education, Manipal, India. ✉email: aalnaser@aus.edu

The trihydrogen cation, $H_3^+$, the simplest and most abundant triatomic ion in the universe, has continued to attract the research interest of numerous scientific communities since its discovery by J. J. Thomson in the early twentieth century[1]. Acting as a catalyst, $H_3^+$ is known to drive interstellar chemical reactions that lead to the creation of dense molecular clouds comprising diverse organic molecules[2,3]. As it behaves like a Brønsted–Lowry acid[4], the trihydrogen cation protonates interstellar ions, atoms, and molecules acting as the critically important precursor in the formation of complex organic compounds, which are believed to play a vital role in the creation of life in the universe[5,6]. It is not surprising then that the processes involved in the creation of $H_3^+$ have been the focus of many scientific investigations. Experiments investigating the formation of $H_3^+$ from organic compounds using electron impact[7,8], highly charged ion (HCI) collisions[9,10], and intense laser fields[11–15] are well documented in the literature. In these studies, usually a single organic molecule undergoes bond cleavage and bond formation in a two-step chemical process resulting in the creation of the trihydrogen cation. The first step involves a doubly charged precursor ion that breaks into a neutral $H_2$ molecule and a doubly charged fragment of the original organic molecule. In the second step of the mechanism, the roaming hydrogen molecule abstracts a proton from the doubly charged fragment, leading to the creation of an $H_3^+$ ion. At variance with all previous studies, in this work, we investigate the formation of the trihydrogen cation from inorganic molecules in a bimolecular mechanism encompassing two $H_2O$ molecules adsorbed on the surfaces of silica nanoparticles in an intense femtosecond laser field.

As a hybrid between the atomic and the macroscopic realms, nanoparticles, and in general nanomaterials, are of increasing utility due to their unique characteristics and potential advantages in many science and engineering fields. For instance, nanoparticles have been utilized in laser acceleration of ions[16,17] and plasma dynamics[18]. In addition, based on their surface chemistry, nanoparticles have been shown to induce unexpected effects on biological tissues and cells. To illustrate, mesoporous silica nanoparticles have shown a remarkable ability as drug delivery agents when used in targeted cancer therapies, opening possibilities of a potentially safer and more effective treatment compared to contemporary methods such as chemotherapy[19]. The promising applications of nanoparticles in various fields are driving the extensive research on nanoscale materials. Specifically, in the light–matter interaction domain, intense femtosecond lasers have been used to study the effects of composition, shape, orientation, and size of nanostructures on their light absorption properties, and the spatial distribution of their photoion[20–22] and photoelectron[23–25] emissions in isolated nanoparticles and their clusters[26]. To further the understanding of the impact of the nanostructures' near-field on emitted photoions, Hickstein et al.[20] demonstrated how the composition of the nanostructures influence the momenta distributions of emitted photoions. Antonsson et al.[21] investigated the emission asymmetry of photoions ejected from NaCl nanoparticles as functions of size and laser intensity. On the other hand, Rupp et al.[22] demonstrated the enhancement in energy and momenta of protons dissociated from molecules inhabiting the surfaces of nanoparticles. Rosenberger et al.[26] highlighted the contrast in the momenta distribution of protons emitted from single nanoparticles compared to dimers. The aforementioned investigations have provided significant insights into the role of near-field enhancement in the photoion and photoelectron emissions from nanoparticles in intense laser fields[22–26]. However, in previous investigations[22,26], the parent atom or molecule of the emitted ions remained unconfirmed. Filling this lacuna would allow tracking and potentially controlling the molecular dynamics that lead to the generation of emitted

ions or electrons and assist in the development of new theoretical models. Here we present results that unravel complex and unique molecular dynamics carried by water/heavy water molecules adsorbed on the surfaces of $SiO_2$ nanoparticles to form $H_3^+/D_3^+$ ions. This bimolecular photochemical reaction, which involves proton/deuteron migration as well as bond cleavage and bond formation, is shown to lead to the formation of $H_3^+/D_3^+$ from two water/heavy water molecules. Our results offer unambiguous demonstration of yet another characteristic of nanoparticles as catalysts for exotic chemical reactions in intense femtosecond laser fields.

## Results

In the present experiments, we employed the reaction nanoscopy technique[22], which enables measuring the three-dimensional momenta and energy of photoions. This experimental technique allows for differentiating between the charged fragments emitted from the nanoparticle surfaces and those generated from background gas constituents. To discriminate the ionization events associated with nanoparticles, where the electron emission signal is much higher compared to the signal from background gas constituents, we used a channeltron electron multiplier to record electron emission in coincidence with the ion emission. Further specifications of the experimental apparatus are schematically illustrated in the "Methods" section. We used a high-power, infrared fiber-based laser system (Active Fiber, central wavelength of 1030 nm) with a repetition rate of 150 kHz, pulse duration of ~40 fs, linearly polarized pulses, and focused intensities in the range of $1 \times 10^{14}$ to $9.5 \times 10^{14}$ W/cm$^2$.

To investigate the formation of $H_3^+/D_3^+$ from water/heavy water molecules adsorbed on the surface of silica nanoparticles, we conducted a series of experiments on $SiO_2$ nanoparticles (300 and 100 nm diameters), which were suspended in deionized $H_2O$ and $D_2O$, prior to their injection into the vacuum chamber. The silica nanoparticles were exposed to laser intensities that are well-below the intensities where metallization is expected to occur and, therefore, the effects of plasmon excitations can be neglected in our current studies[27]. To ensure the accuracy of our experimental results, we confirmed, by Fourier-transform infrared (FTIR) spectroscopy, that our nanoparticles possessed a pure surface, devoid of any hydrocarbons; the surface exclusively comprised silanols (cf. Supplementary Fig. 1 showing FTIR spectra of the nanoparticles used in our experiments). The absence of hydrocarbons from the surfaces of nanoparticles is essential for our experiments, as it allows to precisely and unambiguously identify the source of the emitted trihydrogen and trideuterium ions.

Figure 1a shows typical time-of-flight (TOF) spectra of ions emitted from the surfaces of nanoparticles. Identifying the origin of the aforementioned ions was enabled by using the reaction nanoscopy method detailed in the "Methods" section. The TOF measurements show the formation of $H_3^+$ in addition to $H^+$ and $H_2^+$ ions. In addition, Fig. 1a shows that the formation of $H_3^+$ ions is nanoparticle-size independent. Although our TOF spectrometry succeeds in confirming the creation of the trihydrogen cation from water molecules adsorbed on the nanoparticle surfaces, the exact source of $H_3^+$ remains ambiguous as the silica nanoparticles' surfaces used in our experiments are inhabited by silanol groups in addition to water. As such, the production of $H_3^+$ may be a result of either water, silanol, or both. Therefore, to unambiguously identify the source of $H_3^+$ cations, we performed a different set of measurements with parameters identical to those mentioned above but using an isotopolog of water, namely $D_2O$. Substituting hydrogen by its isotope is a critical step in the process of identifying the exact origin of the trihydrogen cations, as the existence of deuterium is constrained to the heavy water molecules adsorbed on the nanoparticles' surfaces and cannot be

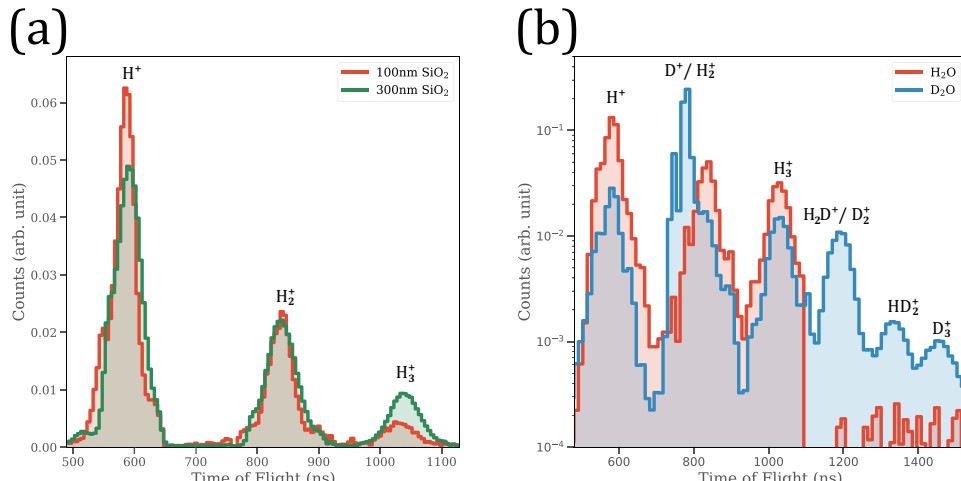

**Fig. 1 Emitted ions from the surfaces of nanoparticles. a** Comparison between two TOF spectra from two experiments (red/green lines) with different conditions including size of nanoparticles (100 and 300 nm), concentration of sample (3 and 1.5 g/L), and laser intensity at $2 \times 10^{14}$ W/cm$^2$ serving as an evidence for the formation of H$_3^+$. **b** Comparison between ions emitted from the surfaces of 100 nm silica nanoparticles inhabited by H$_2$O and D$_2$O molecules at 3 g/L concentration and irradiated by a laser intensity of $2 \times 10^{14}$ W/cm$^2$. The presence of D$_3^+$ and HD$_2^+$ peaks in the ToF spectrum of D$_2$O adsorbed on nanoparticles presents unequivocal evidence about the source of the trideuterium ions (for extended TOF spectra, cf. Supplementary Fig. 4). Source data are provided as a Source Data file.

found in silanol or any other constituents in our vacuum chamber. Figure 1b shows a comparison between the TOF spectra of 100 nm silica nanoparticles suspended in D$_2$O and H$_2$O. The observation of D$_3^+$ when D$_2$O is used as a suspension solution is unequivocal evidence in support of the formation from water adsorbed on the nanoparticle surface. Moreover, the existence of HD$_2^+$ in the TOF spectrum provides further confirmation, as it can only be produced when a D$_2$ moiety is formed through a migration mechanism in a single D$_2$O molecule followed by the abstraction of a proton from a neighboring silanol ion.

The TOF spectra in Fig. 1b reveal a considerable difference in yield between H$_3^+$ and D$_3^+$ emitted from the surfaces of nanospheres inhabited by H$_2$O and D$_2$O, respectively. The relative yield of the trihydrogen ion, which is calculated as the ratio of the area under the H$_3^+$ peak to the sum of areas under H$^+$, H$_2^+$, and H$_3^+$ peaks, was found to be 5.4% when produced from nanoparticle surfaces inhabited by H$_2$O. In the case of D$_2$O, we found the relative yield of D$_3^+$ emitted from D$_2$O molecules on the surface of silica nanoparticles to be ~0.38%. We have also measured the relative yield of HD$_2^+$ to be ~0.54%. Thus, the sum of the yields of D$_3^+$ and HD$_2^+$ remains less than that of H$_3^+$ emitted from the surface of silica nanospheres inhabited by H$_2$O. As isotopologs, H$_2$O and D$_2$O have nearly identical values of ionization energy[28] and a very similar intensity requirement to initiate the hydrogen migration process as well[29,30]. It is therefore expected that the relative yields of H$_3^+$ and D$_3^+$ will be similar. However, the difference in the yields between H$_3^+$ and D$_3^+$ suggests tantalizing possibilities of the existence of multiple novel pathways to form the trihydrogen cation as follows: (i) the production of H$_3^+$ exclusively from water adsorbed on the surface (ii) or solely from silanol, (iii) a hybrid process involving silanol and water wherein H$_2$ is formed from silanol, followed by acquisition of a proton from water, and (iv) a hybrid process where the H$_2$ is generated from water, which then abstracts a proton from silanol. If there were a significant exchange between D$_2$O and SiOH to generate deuterated silanols, D$_3^+$ would have been formed through four different pathways (similar to H$_3^+$ in the case of H$_2$O described above) and, consequently, the TOF spectra in Fig. 1b would have displayed a higher yield of D$_3^+$ compared to H$_3^+$ in the "D$_2$O on SiO$_2$" experiments, as, in this case, the formation of D$_3^+$ would have a pathway from deuterated silanol in addition to two

hybrid pathways and one solely from D$_2$O. However, based on our FTIR spectra taken at different time intervals of the particles in D$_2$O solution (cf. Supplementary Fig. 2) and the observed TOF spectra, we do not find any evidence suggesting significant deuteration of silanols under our experimental conditions.

The unimolecular migration process leading to the creation of H$_2^+$ and HD$^+$ ions from H$_2$O[30] and HOD[29] molecules, respectively, in an intense irradiating field has been documented before. According to Mathur et al.[29], producing the dihydrogen cation requires the application of few-cycle pulses (~7 fs) and laser intensity of ~$1 \times 10^{15}$ W/cm$^2$. Hence, we conducted a series of experiments on D$_2$O in the gas phase, to investigate the possibility of forming of D$_2^+$ ions at intensities, where both D$_2^+$ and D$_3^+$ are generated from D$_2$O molecules on the surfaces of nanoparticles. The TOF spectra from the experiments with gaseous D$_2$O are displayed in Fig. 2a. The results of our measurements did not yield any evidence for the creation of D$_2^+$ cations from D$_2$O in the gas phase, even at the highest intensity of ~$1 \times 10^{15}$ W/cm$^2$. At the same time, an incident intensity of $1 \times 10^{14}$ W/cm$^2$, an order of magnitude lower, was sufficient to produce D$_2^+$ and D$_3^+$ cations from D$_2$O adsorbed on 100 nm SiO$_2$ particles (cf. Fig. 2b).

Based on the experimental evidence, we can conclude that silica nanoparticles act as a catalyst facilitating the formation of the trihydrogen/trideuterium cation from two water/heavy water molecules residing on the surface of the nanoparticles in a bimolecular photoreaction.

In addition to playing a catalyst role in forming H$_3^+$ and D$_3^+$ cations from water, we have discovered that the nanoparticles also facilitate the enhancement of momenta and energy of H$_3^+$/D$_3^+$ fragments formed on the surface of the nanoparticles. The enhancement is attributed to the formation of the near-field around the nanoparticle due to the interaction with a strong laser field, which is in accordance with earlier reports[22,26] on the enhancement of energy and momenta for protons dissociating from the surface of silica nanoparticles. Figure 3 shows the momenta and energy distributions of H$_3^+$ cations. The H$_3^+$ kinetic energy was found to have a mean value ~15 eV confirming the double role of nanoparticles as a facilitator of a bimolecular reaction between two water molecules and a source of enhancement in energy as well. Similarly, the mean energy for D$_3^+$ cations was also found to be ~15 eV (cf. Supplementary Fig. 7b).

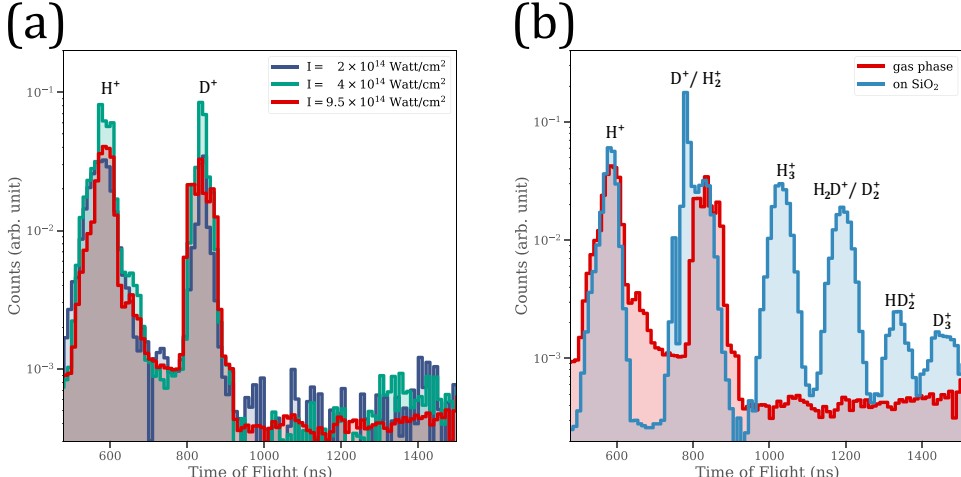

**Fig. 2 Ion emission from $D_2O$ in gas phase and on nanoparticles. a** Comparison between time-of-flight spectra for $D_2O$ in the gas phase for different laser intensities as indicated. The observed $H^+$ ions originate from background gas. **b** Comparison between TOF spectra taken for $D_2O$ in the gas phase at $9.5 \times 10^{14}$ W/cm² laser intensity and ions emitted from the surface of 100 nm nanoparticles inhabited by $D_2O$ molecules at a laser intensity of $1 \times 10^{14}$ W/cm². Only the spectrum associated with nanoparticles demonstrates the formation of $D_2^+$, $HD_2^+$, and $D_3^+$ ions. Source data are provided as a Source Data file.

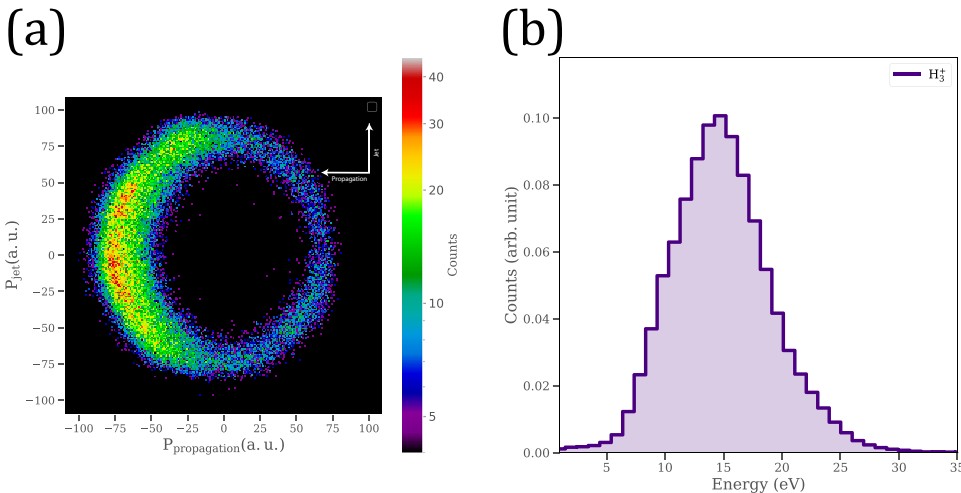

**Fig. 3 Momentum and energy of $H_3^+$. a** 2D momentum image demonstrating the angular distribution in the plane of laser propagation vs. nanoparticles jet where the laser polarization is perpendicular to this plane and **b** energy distribution of $H_3^+$ emitted from the surface of 300 nm silica nanoparticles inhabited by $H_2O$ molecules at a laser intensity of $2 \times 10^{14}$ W/cm². Source data are provided as a Source Data file.

Simulating the enhanced field around single silica nanoparticles[22] and their dimers[26] demonstrated how, for isolated nanoparticles, the near-field reaches its maximum intensity at the polar regions of the nanoparticle along the laser polarization, whereas in the case of a dimer, the near-field reaches its highest magnitude along the propagation direction. The angular distributions of $H_3^+$ (cf. Fig. 3a) and $D_3^+$ (cf. Supplementary Fig. 7a) show that the tri-hydrogen/trideuterium cations formed from water and heavy water, respectively, are mainly emitted from the surfaces of nanoparticles along the laser propagation direction. Such angular distribution is similar to what was previously observed with $H^+$ emitted from 300 nm silica nanoparticle clusters[26].

## Discussion
Quantitative explanation of the processes involved in the formation of the high energy $H_3^+$ and $D_3^+$ ions observed in our experiments defies existing theoretical models. Any theoretical model that is to be developed needs to properly account for the dynamics of laser-field-induced ionization and fragmentation of water molecules on the surface of nanoparticles. In the absence of

such viable theoretical approaches, we take recourse to a qualitative rationalization of our experimental observations by postulating two potential mechanisms that could account for the creation $H_3^+/D_3^+$ from water molecules. These mechanisms are illustrated in Fig. 4. The first formation pathway (cf. green dashed line in Fig. 4) involves the aggregation of water molecules on the surface of $SiO_2$ nanoparticles. It is well known that each H-atom in a water molecule can be weakly attracted to a neighboring O-atom in an adjacent water molecule, leading to the formation of a hydrogen bond. Despite the relatively low energy of the hydrogen bond, estimated at ~250 meV[31], it is still sufficient to form a water dimer that is not destroyed by intramolecular thermal collisions at temperatures lower than 25 °C. The formation of such water clusters is not observed in our gas-phase experiments where our gas flow into the laser interaction zone is subsonic. Therefore, these aggregates must be a consequence of the nanoscale environment provided by the nanoparticles themselves. The creation of $H_3^+/D_3^+$ from water aggregates follows a two-step process involving both hydrogen/deuterium migration and roaming. In the first step, a migration process is initiated by the laser field

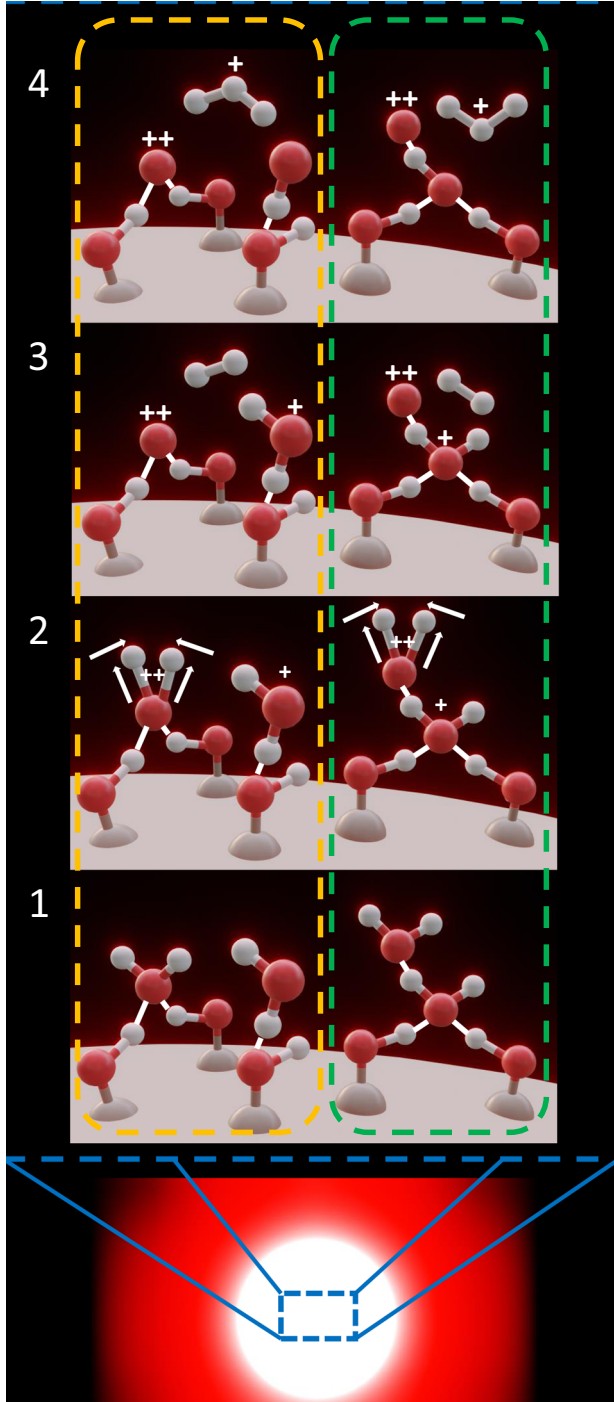

**Fig. 4 Bimolecular reaction to form $H_3^+$ from water molecules on $SiO_2$ nanoparticles.** An illustration of two potential processes responsible from the formation of $H_3^+$ from water molecules on the surface of silica nanoparticles. The bottom panel demonstrates a nanoparticle in an intense laser field. The orange dashed line depicts the formation of $H_3^+$ from two separate water molecules through a roaming on the surface mechanism. The green dashed line depicts the roaming mechanism within a water dimer. In both cases, there exist four stages. Stage 1 shows the ground state. Stage 2 shows the migration mechanism leading to the formation of a hydrogen molecule. Stage 3 is roaming close to another water ion. Stage 4 is the formation of $H_3^+$.

where the distance between the two hydrogen atoms decreases until a neutral $H_2/D_2$ moiety is dissociated from a doubly charged $H_2O^{2+}/D_2O^{2+}$ ion. Second, the newly created hydrogen molecule abstracts a proton from its neighboring water ions as it roams around the water cluster, leading to the creation of an $H_3^+/D_3^+$ cation. The second plausible explanation for the $H_3^+/D_3^+$ production mechanism (orange dashed line in Fig. 4) is similar to what has already been observed and rationalized in unimolecular reactions in alcohol molecules[11–15]. The formation mechanism, similar to the process in clusters, involves both hydrogen migration and roaming. The migration step of the process remains unaffected; however, the roaming part of the procedure is altered in that the freshly produced $H_2/D_2$ molecule does not roam around its water cluster. Instead, the hydrogen molecule roams around the nanoparticle's surface, under the influence of the nanosphere's local field, and abstracts a proton from a neighboring water ion, thereby forming $H_3^+/D_3^+$ in a bimolecular reaction in contrast to a unimolecular reaction in alcohol molecules.

It is well established that when exposed to an intense electromagnetic field, nanoparticles experience an enhanced near-field with a spatial distribution that is different from the incident external beam[20–26]. Furthermore, the local field created around the nanoparticle has a higher intensity compared to the irradiating pulses. This enhancement in intensity differs depending on the nanoparticle's composition[20]. Simulating the field enhancement of $SiO_2$ nanoparticles using the finite-difference time-domain method yields a maximal intensity enhancement of 2.34 and 2.69 for isolated 100 and 300 nm particles, respectively[22], and 6.76 for dimers[26]. Still, the enhanced intensity obtained by using nanoparticles remains lower than the lowest intensity required to induce a migration mechanism in $H_2O$ molecules in the gas phase according to Rajgara et al.[30], which is in accordance with our findings from the gas-phase experiments (cf. Fig. 2). The lower external intensity in our studies is not only adequate for initiating migration within a water molecule but also sufficient to start the complex mechanisms involved in forming the trihydrogen/trideuterium cations from two water/heavy water molecules on the surface of irradiated nanoparticles. Therefore, nanoparticles act as catalysts in the creation of $H_3^+/D_3^+$ by providing a unique nanoscale environment. This nanoscale environment encompasses the near-field intensity enhancement, as well as the trapped electrons[25] and local charges interactions, which can facilitate the opening of such a novel reaction.

To summarize, we have conducted a series of experiments involving intense laser irradiation of water and heavy water molecules adsorbed on the surfaces of silica nanoparticles. Our findings offer unambiguous evidence of $SiO_2$ nanoparticles playing the role of catalysts in the formation of $H_3^+/D_3^+$ with high kinetic energy and surprisingly high yields. The creation of $H_3^+/D_3^+$ requires, of necessity, a bimolecular reaction between two water molecules on the surface of nanoparticles. We have demonstrated the possibility of probing complex molecular reactions occurring on the surface of nanoparticles in an intense laser field. These results should reinvigorate efforts to overcome the profound theoretical challenges that are currently faced in developing proper theoretical insights into molecular dynamics that occur concomitantly in intense optical fields and on the surfaces of nanostructures. There is growing contemporary evidence that the implications of such dynamics may well extend beyond the domain of the chemical sciences and into the biological sciences[32]. In the context of the fact that the existence of $H_n^+$ entities in diverse astrophysical environments has been conjectured for about six decades[33], the experimental results that

we present here may be relevant to astrophysics and astro-chemical environments, and in particular to the evolving field of dust astronomy. Although the environment conducive to the production of trihydrogen cation presented in this work was engineered in our terrestrial laboratory, nature has engineered its own parallel environment. It is now well established that dust in both the inner heliosphere and planetary debris discs around other stars does include nanometer-sized dust particles ranging in size from a few tens to a few hundreds of nanometers[34,35]. Indications of hydrated cometary interplanetary dust particles followed by laboratory studies suggest that hydrated silicates may form in the near-surface regions of comets if liquid water is present[36]. It is not unrealistic, then, to expect these dust nanoparticles to act as catalysts for the formation of trihydrogen cation, or other molecular species requiring similar conditions, if these nanoparticles are impacted by cosmic rays of charged particles or solar wind HCIs.

To further elaborate on this point, it has been established that in collisions with charged particles, molecular targets experience fields equivalent to half-cycle pulses, "virtual photons," of similar magnitudes to those of intense fields of femtosecond, or shorter, laser pulses of several optical cycles[9,37]. The time-dependent effective intensity $I_{eff}(t)$ depends on the impact parameter, charge state, and velocity of the charged particle with peak intensities that can easily exceed $10^{13}$ W/cm$^2$. In addition, the experimental observation of the formation of $H_3^+$ cations in low energy (1.2 MeV $\approx$ 2400 km/s) collisions of Ar$^{8+}$ ions with gas-phase methanol led to the suggestion that collisions of HCI with organic molecules is a possible mechanism for the formation of $H_3^+$ in space[9]. Following Bhardwaj et al.[37], the effective peak intensity for the 2400 km/s Ar$^{8+}$ ions at an impact parameter of 3 Å would be $\sim 3.5 \times 10^{13}$ W/cm$^2$, whereas the effective peak intensity for a fast ($\approx$ 800 km/s) solar wind HCI of similar charge state such as Ne$^{8+}$, Mg$^{8+}$, Si$^{8+}$, or Fe$^{8+}$ ions would be $\sim 1.2 \times 10^{13}$ W/cm$^2$, which is about one-third that of the Ar$^{8+}$ laboratory effective peak intensity. Given that the present results have demonstrated that for D$_2$O adsorbed on SiO$_2$ nanoparticles, the production of D$_2^+$ and D$_3^+$ cations required an incident femtosecond laser intensity one order of magnitude lower than

that needed to produce D$_2^+$ cations from D$_2$O in the gas phase, hydrated dust nanoparticles in space may indeed act as catalysts for the formation of $H_3^+$ cations under the impact of solar wind HCI of somewhat smaller effective peak intensities than that of Ar$^{8+}$. Cosmic rays of charged particles, on the other hand, can give rise to effective peak intensities several orders of magnitude larger than those of solar wind HCI at sub-attosecond timescales and may readily initiate the sequence of events leading to the formation of $H_3^+$ from hydrated dust nanoparticles. In this context, the historical connection between attosecond atomic collisions and ultrashort light pulses has been recently summarized in ref. [38].

## Methods

**Experimental setup.** To conduct the experiments, we employed the reaction nanoscope technique[22] illustrated in Fig. 5. We used a high-power fiber-based laser system (Active Fiber) at a central wavelength of 1030 nm, repetition rate of 150 kHz, pulse durations of around 40 fs, linearly polarized pulses along the electric field, and intensities in the range of $1 \times 10^{14}$ to $9.5 \times 10^{14}$ W/cm$^2$. The laser was tightly focused inside the reaction nanoscope's ultrahigh vacuum chamber using a spherical silver back-focusing mirror. We used two sets of silica nanospheres with a diameter of 300 and 100 nm as targets. To ensure the accuracy of our experimental results, we used nanoparticles with a pure surface, such that it is clean from any hydrocarbons and is inhabited by silanol (SiOH) exclusively. The absence of hydrocarbons from the surfaces of nanoparticles is essential for our experiments, as it allows us to precisely identify the source of the emitted $H_3^+/D_3^+$ ions. These nanoparticles were suspended and sonicated in deionized H$_2$O/D$_2$O and the mixture was aerosolized using a constant output atomizer (TSI 3076). The aerosol is delivered into the ultrahigh vacuum chamber through an aerodynamic lens, creating a jet of aerosolized nanoparticles crossing the laser beam path, at the center of the chamber, resulting in the formation of ion fragments from the H$_2$O/D$_2$O molecules that were adsorbed by the silanol groups on the surface of the nanoparticles. The reaction nanoscopy technique permits three-dimensional momentum imaging of charged fragments, from which their kinetic energies can be obtained. In the experiments, we measure the TOF and the impact position of ions using a delay-line detector equipped with two microchannel plates and a delay-line anode.

**Ions origin identification.** To distinguish ions emitted from the surface of the nanoparticles from those originating in the background gas in the chamber, we utilized a channeltron electron multiplier (Photonis Magnum). The channeltron records the number of electrons associated with each laser shot. Nanoparticles generate a larger number of photoelectrons when interacting with the laser pulses compared to the background gas. This in turn allows the identification of ions

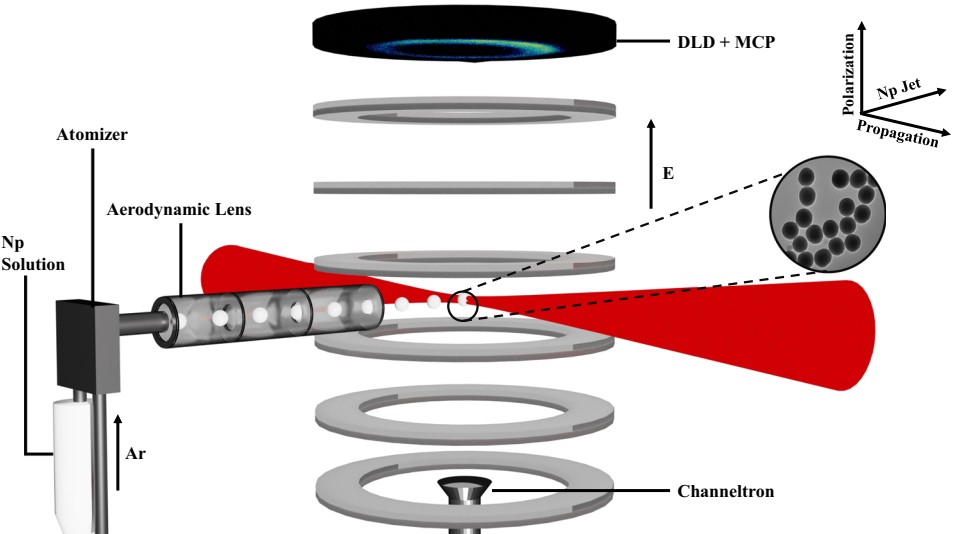

**Fig. 5 Experimental setup.** Illustration of the experimental setup showing the nanoparticle solution (Np Solution) that is aerosolized by an atomizer with the help of Ar as carrier gas. The aerosol is then collimated using the aerodynamic lens producing a jet that is admitted to the ultrahigh vacuum chamber. The nanoparticles interact with the laser field in the center of a constant-field (E) spectrometer. The ions generated from the interaction are accelerated towards a microchannel plate (MCP) and delay-line detector (DLD). As for the electrons, they are accelerated towards a channeltron and are used to discriminate between nanoparticles and background ions. The inset shows a scanning electron microscope (SEM) image of the silica nanoparticles (NanoComposix) used in the experiments.

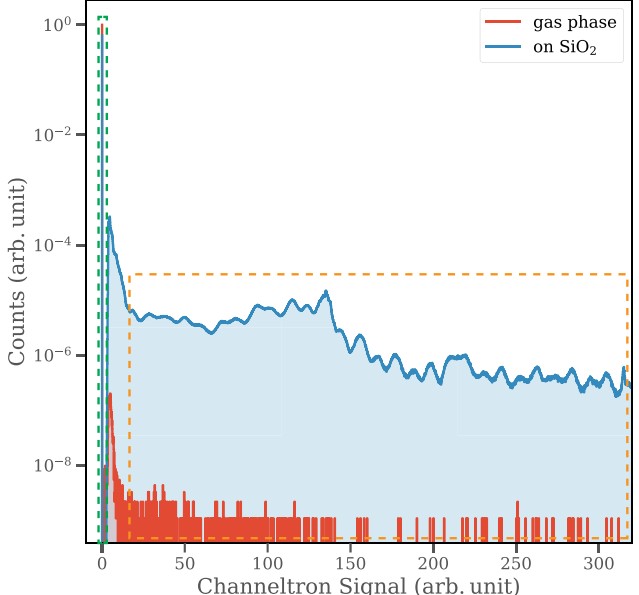

**Fig. 6 Channeltron signal for gas-phase and nanoparticle targets.** Figure 6 displays accumulated distributions from ~1 billion pulses in the experiment. A considerable difference in channeltron signal between gas phase and nanoparticles $D_2O$ targets. The red line represents the channeltron signal for $D_2O$ in the gas phase, whereas the blue curve demonstrates the signal associated with nanoparticles. Both measurements were performed at a laser intensity of $2 \times 10^{14}$ W/cm². The vast majority of laser shots result in no channeltron signal as indicated by the black dashed line. The orange dashed line highlights how laser shots associated with irradiating nanoparticles generate a significantly larger channeltron signal, which in turn is used in coincidence with the time-of-flight spectra to extract the time of flight for ions associated with nanoparticles. Source data are provided as a Source Data file.

associated with nanoparticles only by filtering the TOF signal measured in coincidence with a high channeltron signal. Figure 6 demonstrates the considerable difference in channeltron signal between nanoparticles and the background gas constituents.

## Data availability

The data that support the findings of this study are available from the corresponding author (A.S.A.) upon reasonable request. Source data are provided with this paper.

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

## Acknowledgements

We acknowledge support from the American University of Sharjah through the FRG_19_LS61 Grant and the AUS-Common Research Facility. R.A. acknowledges sabbatical leave granted by the University of Jordan during the 2019/2020 academic year. We acknowledge support by the Max Planck Society via the Max Planck Fellow program and the International Max Planck School of Advanced Photon Science (IMPRS-APS). We are grateful for support by the German Research Foundation (DFG) via LMUexcellent and project KL-1439/11-1. We thank Dr. Hussain Alawadhi from the University of Sharjah for help with the FTIR measurements.

## Author contributions

A.S.A. conceived and supervised the study. M.S.A. and R.A. prepared and performed the experiments. M.S.A. analyzed the data. A.S.A. and M.F.K. introduced the nanoTRIMS concept. M.S.A., R.A., P. Rupp, P. Rosenberger, S.M., and R.D. set up the experimental apparatus. V.K. and M.I. provided the laser beam and assisted in performing the measurements. Final interpretations were carried out by M.S.A., R.A., A.S.A., M.F.K., P. Rosenberger, and B.B. with contributions from other authors. Many valuable insights and suggestions were provided by D.M. All authors discussed the results and contributed to the final manuscript drafted by M.S.A. and A.S.A.

## Competing interests

The authors declare no competing interests.
