## [Peer Review File · Nature Communications]

REVIEWER COMMENTS

Reviewer #1 (Remarks to the Author):

The manuscript presents evidence for the formation of trihydrogen cations from water on silica nanoparticles. I find the manuscript to be very well written. The introduction is very clear, and the results are very clear. The authors present sufficient control experiments in their paper and the supporting material to justify their assignments and claims.

This is an exciting first observation that warrants further experiments and theory. For example, one would like to know if trihydrogen cations would form on wet nanoparticles not containing H atoms. Similarly, experiments on water droplets would inform the need for the nanoparticles.

I am sure the authors will be addressing these and many other questions in the future.

Reviewer #2 (Remarks to the Author):

The authors report their novel study of H_3^+ formation from water on silica nanoparticles, following excitation by intense femtosecond laser pulses, using mass-resolved ion momentum imaging, with electron detection in coincidence. The experiments reveal a rich level of detail to access the mechanisms of H_3^+ (D_3^+) formation in silica nanoparticles prepared with H_2O (D_2O), which have profound implications on the inorganic synthesis of this highly important ion in aerosols and in space. The study is highly likely to appeal to a broad readership and will likely inspire follow-on work in this field. The technical information in the manuscript and Supplementary Information is clearly presented, and I expect an expert researcher could reproduce the work using the provided information. Overall, the manuscript is a very clear and concise account of challenging and highly significant research, so it is appropriate for Nature Communications. There are, however, a few issues with the manuscript that should be addressed before further consideration.

The main conclusions are that H_3^+ is formed in a bimolecular reaction between two water molecules in the presence of the silica nanoparticle surface. Although a great deal of ambiguity is removed by the careful sample preparation devoid of hydrocarbons, still some ambiguity seems to remain. It is

well-established that deuterium can exchange with hydrogen in solutions and aerosols of silanol and water (see for example Bakaev and Pantano, *J. Phys. Chem. C* 2009, 113, 31, 13894–13898 <https://doi.org/10.1021/jp810984r>). Therefore, a significant concentration of silanols on the nanoparticle surface may be deuterated in the D₂O experiments. Can the authors rule out or quantify the presence of deuterated silanols? The possible bimolecular reactions of deuterated silanols (pathway ii in the manuscript) and hybrid pathways (iii and iv) involving deuterated silanols and D₂O to produce each of the triatomic isotopologues H₂D⁺, HD₂⁺ and D₃⁺ should be addressed in the revised manuscript.

I also have a technical question regarding the electron coincidence signals that determine the presence of one or more nanoparticles. In the laser focus, the background gas is expected to emit a much lower yield of photoelectrons compared to one or more nanoparticles, therefore high and/or broad pulses from the channeltron detector correspond to multiple-ionization of a nanoparticle. It is unclear whether Figure S2 shows a single-pulse analysis, or an accumulated pulse height distribution over many laser shots. The “gas phase” pulse height distribution is a factor of about 500 smaller than the “on SiO₂” distribution, but otherwise the two are qualitatively similar. The authors should clarify how many laser pulses contribute to each plot in Figure S2.

Reviewer #3 (Remarks to the Author):

This communication reports interesting results on the formation of H₃⁺ on the surface of water covered silica nanoparticles exposed to intense, femtosecond laser pulses. By comparison of surface and gas phase experiments and the use of D₂O background effects are stated to be excluded. The creation of H₃⁺ is attributed to a catalytic cycle process on the surface involving both bond fragmentation and diffusion of water molecules on the surface. This is an interesting result but the authors should quantify a few points.

1. Did they collect a FTIR spectra of the water coated silica nanoparticles as they did to observe absence of organics?

2. What was the background pressure in the vacuum chamber? Was a residual gas analysis of the vacuum recorded? If so what were levels of background water and from these how long would it take to form a layer of adsorbed water on the nanoparticles (e.g. to mix with the D₂O)?

3. Photon interactions with nanoparticles may invoke plasmon excitations – are these excluded in the present case?

The authors seek to make a comparison with ISM studies and creation of H_3^+ on the surface of water coated ISM particles. However the experimental conditions they are using with femtosecond lasers is far removed from any photon fluxes in the ISM or other planetary regions. The authors should therefore clarify how they postulate that the present experiments can be compared with astrochemical conditions.

Reviewers' comments:

Reviewer #1:

- The manuscript presents evidence for the formation of trihydrogen cations from water on silica nanoparticles. I find the manuscript to be very well written. The introduction is very clear, and the results are very clear. The authors present sufficient control experiments in their paper and the supporting material to justify their assignments and claims.

This is an exciting first observation that warrants further experiments and theory. For example, one would like to know if trihydrogen cations would form on wet nanoparticles not containing H atoms. Similarly, experiments on water droplets would inform the need for the nanoparticles.

I am sure the authors will be addressing these and many other questions in the future.

Reply: We thank the reviewer for the positive remarks on our work and for finding our manuscript worth publication in Nature Communications. We also appreciate the useful suggestion on exploring the formation of trihydrogen cations from wet nanoparticles not containing H atoms, which will be indeed a target in our future studies, as the reviewer suggested.

Reviewer # 2:

The authors report their novel study of H₃⁺ formation from water on silica nanoparticles, following excitation by intense femtosecond laser pulses, using mass-resolved ion momentum imaging, with electron detection in coincidence. The experiments reveal a rich level of detail to access the mechanisms of H₃⁺ (D₃⁺) formation in silica nanoparticles prepared with H₂O (D₂O), which have profound implications on the inorganic synthesis of this highly important ion in aerosols and in space. The study is highly likely to appeal to a broad readership and will likely inspire follow-on work in this field. The technical information in the manuscript and Supplementary Information is clearly presented, and I expect an expert researcher could reproduce the work using the provided information. Overall, the manuscript is a very clear and concise account of challenging and highly significant research, so it is appropriate for Nature Communications. There are, however, a few issues with the manuscript that should be addressed before further consideration.

Reply: We thank the reviewer for the positive remarks on our work and its implications and for recognizing the significance and technical challenges of the presented research. We also thank the reviewer for finding our manuscript worth publication in Nature Communications. Below we provide a point-by-point response to the questions and comments raised by the reviewer:

Comment 1: The main conclusions are that H₃⁺ is formed in a bimolecular reaction between two water molecules in the presence of the silica nanoparticle surface. Although a great deal of ambiguity is removed by the careful sample preparation devoid of hydrocarbons, still some ambiguity seems to remain. It is well-established that deuterium can exchange with hydrogen in solutions and aerosols of silanol and water (see for example Bakaev and Pantano, J. Phys. Chem. C 2009, 113, 31, 13894–13898 <https://doi.org/10.1021/jp810984r>). Therefore, a significant concentration of silanols on the nanoparticle surface may be deuterated in the D₂O experiments. Can the authors rule out or quantify the presence of deuterated silanols? The possible bimolecular reactions of deuterated silanols (pathway ii in the manuscript) and hybrid pathways (iii and iv) involving deuterated silanols and D₂O to produce each of the triatomic isotopologues H₂D⁺, HD₂⁺ and D₃⁺ should be addressed in the revised manuscript.

Reply: We thank the reviewer for pointing us to the reference in J. Phys. Chem, in which GC/MS chromatography was employed to study the (H-D) exchange between silanol groups on the surface of fumed silica, and deuterium oxide in the carrier gas. The process in the article, which is shown to take place over many minutes, involves heating the fumed silica column up to 100° C for the H-D exchange to take a place and at relatively high pressure (~10⁻⁷ Torr) of D₂O gas carrier. The experimental setup also contains an MSD and involves electron impact ionization process to decompose D₂O. In our case, all our experiments were conducted at significantly lower temperature (room temperature) and in a vacuum chamber where the base background pressure was three orders of magnitude lower (in the high 10⁻¹⁰ mbar range). Additionally, we have collected FTIR spectra (please see Fig. R1 below, which is now added as Fig. S4 in the Supplementary Information file) of the nanoparticles samples at different time intervals from the time when they were immersed in D₂O. Our spectra did not exhibit any change overtime and showed no indication of detectable H-D exchange between D₂O and SiOH, even over the course of 22 hrs.

Furthermore, we note that, if there were a significant exchange between D₂O and SiOH, D₃⁺ would have been formed through 4 different pathways (similar to H₃⁺ in the case of H₂O as described in the manuscript), and consequently the time of flight spectra in Fig. 1b and Fig. 2b would have displayed a higher yield of D₃⁺ compared to H₃⁺ in the “D₂O on SiO₂” experiments. Because in this case, the formation of D₃⁺ would have a pathway from deuterated silanol in addition to 2 hybrid pathways and 1 solely from D₂O. However, based on our FTIR spectra and the observed time of flight spectra, we do not find any evidence to suggest significant deuteration of silanols under our experimental conditions. As suggested by the reviewer, we

have now added on page 6 of the revised manuscript a discussion of the possible reaction pathways involving deuterated silanols.

Fig. R1. Typical FTIR spectra of the silica nanoparticle immersed in D₂O and collected at different time intervals from the moment the particles were immersed (T= 0 hrs) until 22 hours (T = 22 hrs) after immersion. The spectra did not exhibit any change overtime and showed no indication of detectable H-D exchange between D₂O and SiOH, even over the course of 22 hrs.

Comment 2: I also have a technical question regarding the electron coincidence signals that determine the presence of one or more nanoparticles. In the laser focus, the background gas is expected to emit a much lower yield of photoelectrons compared to one or more nanoparticles, therefore high and/or broad pulses from the channeltron detector correspond to multiple-ionization of a nanoparticle. It is unclear whether Figure S2 shows a single-pulse analysis, or an accumulated pulse height distribution over many laser shots. The “gas phase” pulse height distribution is a factor of about 500 smaller than the “on SiO₂” distribution, but otherwise the two are qualitatively similar. The authors should clarify how many laser pulses contribute to each plot in Figure S2.

Reply: Figure S2 displays an accumulated distribution over all laser pulses in an experiment. In the presented figure, the plot labeled “on SiO₂”, which corresponds to the measurement where the target is D₂O adsorbed on SiO₂ nanoparticles, represents around 1.2 billion laser pulses. On the other hand, the plot labeled “gas phase”, representing the measurement where the target is D₂O in the gas phase, corresponds to around 1 billion laser pulses. As for their distributions, the measurements where SiO₂ nanoparticles are present exhibit a clear high channeltron signal which corresponds to a large number of electrons emitted as a result of ionizing nanoparticles. However, in the case of D₂O in the gas phase, the distribution of the channeltron signal beyond 10 (arb. unit) occurs at random and is featureless compared to the signal from nanoparticles. Moreover, the “gas phase” signal is in reality a factor of 1.2 smaller than the “on SiO₂” one (as calculated from the area under the curves), which suggests that even qualitatively there is a clear distinction between the two distributions. As suggested by the reviewer, the number of laser pulses contributing to each plot in Figure S2 is now explicitly mentioned in the modified caption of Figure S2 in the revised supplementary information file.

Reviewer # 3:

This communication reports interesting results on the formation of H₃⁺ on the surface of water covered silica nanoparticles exposed to intense, femtosecond laser pulses. By comparison of surface and as phase experiments and the use of D₂O background effects are stated to be excluded. The creation of H₃⁺ is attributed to a catalytic style process on the surface involving both bond fragmentation and diffusion of water molecules on the surface. This is an interesting result but the authors should quantify a few points.

Reply: We thank the reviewer for the positive remarks on our work and for finding our results interesting. Below we provide a point-by-point response to the specific questions and comments raised by the reviewer:

Comment 1: Did they collect a FTIR spectra of the water coated silica nanoparticles as they did to observe absence of organics?

Reply: Indeed we did collect FTIR spectra of the silica nanoparticles suspended in H₂O and the collected spectra confirm the absence of organic molecules from our studied samples. A typical FTIR spectrum of the silica nanoparticles suspended in H₂O is shown in Fig. R2. While those spectra show the different stretching and bending bands for SiOH, SiOSi, HOH and OH, none of the spectra shows any signature of the IR modes in the regions 2800-3000 and 1100-1750 cm⁻¹, which could be associated with any organic attachments to the surface of the nanoparticles.

Fig. R2. Spectra for 100nm SiO₂ powder nanoparticles and the same nanoparticles suspended in H₂O. While those spectra show the different stretching and bending bands for SiOH, SiOSi, HOH and OH, none of the spectra shows any signature of the IR modes in the regions 2800-3000 and 1100-1750 cm⁻¹, which could be associated to any organic attachments to the surface of the nanoparticles. .

Comment 2: What was the background pressure in the vacuum chamber? Was a residual gas analysis of the vacuum recorded? If so what were levels of background water and from these how long would it take to form a layer of absorbed water on the nanoparticles (e.g. to mix with the D₂O)?

Reply: The background pressure inside the chamber is in the 10^{-10} mbar range. A recording of the residual gas spectrum (shown in Fig. R3 below) inside the vacuum chamber reveals that D₂O is the most dominant molecule in the background. The very low density of H₂O in the background inhibits the possibility of forming an additional water layer on the surface of the free-flying nanoparticles. Our experimental apparatus is designed to continuously supply the interaction region with a fast-flying jet of fresh nanoparticles, surrounded by the injection liquid (D₂O), through the aerodynamic lens (cf. figure S1). For our geometry and an estimated speed of 200 m/s of the nanoparticles beam, the nanoparticles spend only a fraction of a millisecond while travelling the few centimeters before reaching the interaction region. As such, the small interaction volume and the very low H₂O density in the chamber reduces the possibility of H₂O adsorbance on the surface of the silica nanoparticles that are surrounded by D₂O. Similar conditions hold, when injecting with the H₂O solvent: we have shown in the manuscript (Figure 1) the time of flight spectra from nanoparticles suspended in H₂O, and the triatomic isotopologues H₂D⁺, HD₂⁺ and D₃⁺ were absent in these spectra.

Fig. R3. Time of flight spectrum of the residual background gas. The spectrum reveals that D₂O is the most dominant molecule in the background, with very low density of H₂O, which inhibits the possibility of forming an additional water layer from the background on the surface of the free-flying nanoparticles.

Comment 3: Photon interactions with nanoparticles may invoke plasmon excitations – are these excluded in the present case?

Reply: Plasmon excitations in dielectric nanoparticles may only occur as a result of their metallization when those particles are exposed to laser pulses of high peak intensities above 2×10^{14} W/cm² (please refer to reference 30 that has been added in the revised manuscript). However, our experiments with nanoparticles were conducted within a range of laser intensities (Fig. 1b and Fig. 2b) that are well-below the intensities where metallization is expected to occur, and therefore, the effects of plasmon excitations

can be neglected in the current studies. To clarify this point, we have added on page 3 of the revised manuscript a statement about the plasmon excitations and why they are neglected in our study. We have also added a new reference (# 30) in the revised manuscript.

Comment 4: The authors seek to make a comparison with ISM studies and creation of H₃⁺ on the surface of water coated ISM particles. However the experimental conditions they are using with femtosecond lasers is far removed from any photon fluxes in the ISM or other planetary regions. The authors should therefore clarify how they postulate that the present experiments can be compared with astrochemical conditions.

Reply: We fully agree with reviewer that conditions we are using with femtosecond lasers are far removed from any photon fluxes in the ISM or other planetary regions, and we have not actually made any such claim. In fact, reference [10] cited in the manuscript clearly states that intense laser fields are not the cause of formation of H₃⁺ in space. The parallel naturally-occurring environment in space we are referring to in the manuscript is the hydrated dust nanoparticles impacted by cosmic rays or solar wind ions, not by intense laser pulses.

The equivalence of the impact of charged particles and ultrashort intense laser pulses has long been addressed in the literature and has been mentioned in reference [10]. To elaborate more on this point and its relevance to our conjecture that our present results are of relevance to astrophysics and astrochemical environments, and in particular to the evolving field of dust astronomy, we have added the relevant, and we believe sufficient, below discussion at the end of the revised manuscript, along with two new relevant references [37, 38] :

“To further elaborate on this point, it has been established that in collisions with charged particles, molecular targets experience fields equivalent to half-cycle pulses, “virtual photons,” of similar magnitudes to those of intense fields of femtosecond, or shorter, laser pulses of several optical cycles^{10,37}. The time-dependent effective intensity $I_{\text{eff}}(t)$ depends on the impact parameter, charge state and velocity of the charged particle with peak intensities that can easily exceed 10^{13} W/cm². In addition, the experimental observation of the formation of H₃⁺ cations in low energy (1.2 MeV \approx 2400 km/s) collisions of Ar⁸⁺ ions with gas-phase methanol led to the suggestion that collisions of HCl with organic molecules is a possible mechanism for the formation of H₃⁺ in space¹⁰. Following Bhardwaj et al.³⁷, the effective peak intensity for the 2400 km/s Ar⁸⁺ ions at an impact parameter of 3 Å would be $\sim 3.5 \times 10^{13}$ W/cm², whereas the effective peak intensity for a fast (\approx 800 km/s) solar wind HCl of similar charge state such as Ne⁸⁺, Mg⁸⁺, Si⁸⁺ or Fe⁸⁺ ions would be $\sim 1.2 \times 10^{13}$ W/cm² which is about one third that of the Ar⁸⁺ laboratory effective peak intensity. Given that the present results have demonstrated that for D₂O adsorbed on SiO₂ nanoparticles the production of D₂⁺ and D₃⁺ cations required an incident femtosecond laser intensity one order of magnitude lower than that needed to produce D₂⁺ cations from D₂O in the gas-phase, hydrated dust nanoparticles in space may indeed act as catalysts for the formation of H₃⁺ cations under impact of solar wind HCl of somewhat smaller effective peak intensities than that of Ar⁸⁺. Cosmic rays of charged particles on the other hand can give rise to effective peak intensities several orders of magnitude larger than those of solar wind HCl at sub-attosecond timescales and may readily initiate the sequence of events leading to the formation of H₃⁺ from hydrated dust nanoparticles. In this context, the historical connection between attosecond atomic collisions and ultrashort light pulses has been recently summarized in reference 38.”

We have also replaced “cosmic rays” with “cosmic rays of charged particles” in the abstract and at the end of the manuscript to avoid any possible confusion.

REVIEWERS' COMMENTS

Reviewer #2 (Remarks to the Author):

In their revised manuscript, the authors have addressed each of the questions raised by each reviewer, and I see no further areas needing refinement. The revised manuscript is a very clear and concise report of important results from very challenging experiments. My recommendation is for the manuscript to be published in Nature Communications without further revision.

Daniel S. Slaughter

Reviewer #3 (Remarks to the Author):

The authors have made a point by point reply to questions and made additions/alterations to the text which are (in respect to my questions and comments) entirely satisfactory.

Reviewers' comments:

Reviewer #2 (Remarks to the Author):

In their revised manuscript, the authors have addressed each of the questions raised by each reviewer, and I see no further areas needing refinement. The revised manuscript is a very clear and concise report of important results from very challenging experiments. My recommendation is for the manuscript to be published in Nature Communications without further revision.

Reply: We thank the reviewer for the very positive remarks on our work and for recommending for publication as is.

Reviewer #3 (Remarks to the Author):

The authors have made a point by point reply to questions and made additions/alterations to the text which are (in respect to my questions and comments) entirely satisfactory.

Reply: We thank the reviewer for the constructive remarks and for finding our response entirely satisfactory.